# Waste, Exclusion, and the Responsibility of the Rich: A Franciscan Critique of Early Capitalist Europe

Dana Bultman

Department of Romance Languages, University of Georgia, Athens, GA 30602, USA; dbultman@uga.edu

**Abstract:** Francisco de Osuna's *Fifth Spiritual Alphabet* of 1542, subtitled *Consolation for the Poor and Warning for the Rich*, is a Spanish text on economic inequality in Western Europe. Osuna treats the life-threatening political divisions of his day, including those intended to reduce people to objects of uselessness and slavery, with spiritual and practical advice that defines free will as true wealth and focuses on the responsibility of the rich for producing poverty. I examine Osuna's theology and sensorial, embodied imagery in dialogue with Francis' encyclical *Fratelli Tutti: On Fraternity and Social Friendship* (2020), particularly Francis' concept of a "beautiful polyhedral reality", through the lens of twenty-first century decolonial feminist and social theory. I argue that Osuna's work is a compelling Franciscan precedent for combating avarice and indifference that is best understood through scholarly perspectives critical of the legacies of patriarchy and colonialism.

**Keywords:** capitalism; decolonial feminism; Franciscans; Francisco de Osuna; *Fratelli tutti*; free will; mercy; patriarchy; Spain

## 1. Introduction

Five centuries ago, the Spanish Franciscan author Francisco de Osuna (c.1492–c.1541) made vivid observations of emergent Western capitalism in his work *Consolation for the Poor and Warning for the Rich* (1542).[1] After witnessing European finance, warfare, slavery, and property practices from his vantage point in the major seaports and trade centers of Antwerp and Seville, Osuna discussed who, and what, he believed was responsible for conflict and poverty. Osuna also gave an audience of Christian laymen and clergymen strategies for bridging divisions within their hearts and across social ranks. Published as volume five of his six-part *Spiritual Alphabet*, Osuna's lengthy and dense text of 221 folios uses evocative sensorial imagery as a persuasive medium for his core messages: he advises his male readers to imagine material wealth as both manna and ripe manure to be swiftly distributed rather than hoarded and left to rot; to picture Jacob's dream ladder, with its rising rails and distinct steps, as a circuit through which generosity naturally flows; and to envision the faithful as a woman, like Moses' Ethiopian wife or the dark-skinned beloved in the Song of Solomon, worthy and capable like Queen Esther (Osuna [1542] 2002, pp. 736, 832, 725–43, 274–86).

To counsel his readers, Osuna uses images of the Sun, the Moon, Earth's elements, and animals to argue that the corrupt practices of powerful men are the principal cause of poverty (ibid., pp. 689–90, 713–21, 737). Therefore, to alleviate suffering one must combat one's own greed and stop robbing and condemning others to exclusion and death. True wealth is free will and an undivided heart, in which one's reason and volition are in harmony, he argues, and the spiritual purpose of material wealth and laws is to lovingly foster life and peace.

In this essay, I examine the natural, biblical, human, racial, and gendered imagery Osuna uses to transmit spiritual teachings and practical advice for action in his text. My method is to read the book comprehensively, and as a whole, to place his arguments in dialogue with Francis' encyclical *Fratelli Tutti: On Fraternity and Social Friendship* (2020),

and to interpret the juxtaposition through a framework of twenty-first century decolonial feminist and social theory. In doing so, I intend, first of all, to increase access to this significant and little-studied text within the Franciscan intellectual tradition, since Osuna is primarily known in the English-speaking world for his influential teachings on the meditative practice of recollection in his *Third Spiritual Alphabet* of 1527.[2] I engage with the work of Spanish scholars, especially Pérez García (2019a, 2019b) and Quirós García (2000, 2002), who have done the most in recent years to interpret the fifth part of Osuna's *Spiritual Alphabet*. Secondly, my deeper purpose, in keeping with the editors' intentions for this special issue, is to study Osuna's relevance to the problem of life-threatening political divisiveness today. My aim is to seek nodes of intersection between his sixteenth-century Franciscan perspectives and current ones, while also taking stock of the patriarchal and supremacist structures in which these perspectives are embedded.

This essay begins by contextualizing Osuna and his work, then presents passages from *Consolation for the Poor and Warning for the Rich* selected to display the principal imagery he transmits. It ends by interpreting the relevance to our current moment of Osuna's critique of economic and social practices in the early decades of capitalist expansion, using *Fratelli Tutti* as a textual interlocutor on themes of generosity and social cohesion. Francis' discussions in *Fratelli Tutti* of combating greed, indifference, and delusion have clear parallels with Osuna's thought. *Fratelli Tutti*'s imagery is also resonant with *Consolation for the Poor and Warning for the Rich*, I argue, particularly for how the encyclical features the figure of the polyhedron to imagine social cohesiveness and mutual respect. In the spirit of the call in *Fratelli tutti* to cross borders and boundaries in order to engage in reciprocal encounter and listening (Francis 2020, § 217), I bring together areas of study that typically operate in separate silos to consider what the substance and tenor of Osuna's complex text can tell us today.

## 2. The Overrepresentation of Western Man and the Polyhedron

As a scholar of Spain's early modern culture working from within the U.S. academy, I am not indifferent to the deep violence initiated in the sixteenth century by the Spanish Empire. Osuna was immersed in that empire and addressed the interests of lay merchants while Renaissance humanism and commercial capitalism were on the rise (Pérez García 2019a). A key legacy of Renaissance humanism, as Wynter (2003) has argued, is the overrepresentation of Western Man. This overrepresentation underpins the injustice of the world we live in today, erroneously projecting "its well-being as that of the human species as a whole, rather than as it is veridically: that of the Western and westernized (or conversely) global middle classes" (Wynter 2003, p. 313). Wynter states her position:

> The argument proposes that the struggle of our new millennium will be one between the ongoing imperative of securing the well-being of our present ethnoclass (i.e., Western bourgeois) conception of the human, Man, which overrepresents itself as if it were the human itself, and that of securing the well-being, and therefore the full cognitive and behavioral autonomy of the human species itself/ourselves. (ibid., p. 260)

Following Wynter, I am guided by long-standing and vigorous efforts to break open racialized, exclusionary ideologies of social relationships that marginalize vast groups of human beings and treat people as if they were expendable waste. As Francis affirms "No one is useless and no one is expendable", and those who are excluded "see aspects of reality that are invisible to the centres of power where weighty decisions are made" (Francis 2020, § 215).

Imagery is my starting point because of the pervasive awareness in our day of the power of images and words to give rise to collective thought and action. Currently, voices from diverse camps are dismantling widely disseminated Western concepts of human nature and society from the past century while redrawing new ones, in an effort to re-create collective relationships for the future. Francis makes such an effort at influencing perceptions with the image of the world as polyhedron. This is the geometric shape he

develops in *Fratelli Tutti* to represent a desirable future: a many-faced composite world made up of differing modes of being human (Francis 2020, § 145, 190, 215). He uses the figure of the polyhedron, a multiplex of equally dignified neighboring cultures and communities, to supplant the ubiquitous image promoted during globalization: the world as a smooth sphere crisscrossed by airplanes and digital connectivity. Unlike a simple circle, or a map of Western economic expansionism, the textured world of the polyhedron is grounded on Earth and embodied. At the edges where the faces meet, Francis envisions "an exchange of gifts for the common good" that will create "a beautiful polyhedral reality" in which everyone has a place (ibid., § 190).

Scholars are similarly working to decolonize histories, center women's experiences, dispel distortions and delusions, and reframe ways to compose a shared future. Kate Raworth's *Doughnut Economics* (2017) and David Graeber's and David Wengrow's *The Dawn of Everything* (2021) creatively replace modernist economic graphs and anthropological presuppositions that, over the past century, have stunted descriptions of human nature and fettered theories of economic and political policy. Tiffany Lethabo King's *The Black Shoals* (2019) returns to the year 1441, well before the rise of Enlightenment thought, when the Portuguese began the systematic introduction of enslaved Black Africans to Europe. Her work offers paths for the collaborative dismantling, by Indigenous peoples and people of color, of the colonial conceptual structures of imperial conquest. Zakiyyah Jackson's *Becoming Human* takes apart the delusions that Western powers produced to represent "black(ened) femaleness and/or femininity" (Jackson 2020, p. 83) as the lowest rung of humanity and thereby enable dehumanization and genocide. Recent scholarship provides models for repudiating concepts of people as "fungible" objects to be used, discarded, or erased (King 2019, pp. 175–206), transforming thought so we can better relate with one another.[3] I consider Osuna's voice in this light.

### 3. Osuna's Life and Overview of Consolation for the Poor and Warning for the Rich

*3.1. Biographical Sketch*

A commoner, born around the year 1492 to parents in service to the estate of the Counts of Ureña in the southern Spanish region of Seville, Osuna rose as a member of the clergy within the Order of the Friars Minor of the Observance by merit of his studies, work, and publications (Quirós García 2002, pp. 27–41). Crafting his books in the environment of Reformation-era religious conflict and the threat of inquisitorial investigations, Osuna deployed a style of copious and circuitous writing. His texts abound in allegorical interpretation of scripture and constellations of nested metaphors delivered with strategic modulations of voice. Alternately plain-spoken, emotive, candid, compassionate, stern, and ironic, by the 1530s Osuna was a well-known Franciscan spiritual author educated in the Scholastic tradition, sensitive to the daily life of the populace, and critical of faults he found in common men and women, his fellow friars, Catholic clergy, merchants, nobles, and contemporary kings (Andrés 1975, pp. 3–117). Unlike his most famous reader, Teresa of Avila, he successfully avoided the Inquisition despite his outspokenness (Bultman 2007, p. 34).

After he gained fame—and attracted controversy—as a teacher of the Franciscan practice of recollection, Osuna diplomatically turned down a post that would have sent him off to Spain's overseas colonial empire in the Americas (Quirós García 2002, p. 37).[4] Instead, he preferred to journey closer to the nerve center of cultural exchange within Western Europe, making his way north through France, on foot, after attending the General Chapter meeting of the Franciscan Order in Toulouse in 1532, to live and work in Flanders, today modern Belgium (Pérez García 2019a, p. 221). Antwerp in the 1530s was home to thousands of Portuguese and Spanish merchants with whom, as we will see, Osuna discussed the legitimacy of emerging financial and trade practices (Braudel 1986, pp. 150–53). Near the end of 1536, he made his return to Spain's northwestern coast and continued to write, despite ill health, until his death around 1541. Perhaps in keeping with his lifelong observation of humility,

no precise details have survived regarding the circumstances of his death or place of burial (Quirós García 2002, p. 41).

*3.2. Osuna as Author in the Context of the Spanish Empire*

Osuna wrote a total of eight spiritual books in Spanish and five in Latin as a confessor and preacher concerned with pastoral care. These were read widely in Europe and owned by Franciscans in the Americas (Bultman 2018, p. 297). The fifth part of his *Spiritual Alphabet*, subtitled *Consolation for the Poor and Warning for the Rich* first appeared in print in Burgos, Spain in 1542 at the expense of the bookshop owner and editor Juan de Espinosa, who did not explain in his admiring prologue to the work how the late friar's manuscript came into his hands. The same publisher, Juan de la Junta, printed a second edition in 1554. The book was also translated into German and published in 1602 in Munich, with the title *Trost der armen und Warnung der Reichen*, and re-edited there again in 1603. It had no further editions until 2002, when Quirós published a much-needed and meticulously prepared critical edition of the Spanish original with extensive notes and a scholarly introduction, which is the version of the text I use here. Currently, there is no English translation.

Firmly based in his Scholastic training, Osuna draws from the Old and New Testaments, the Church Fathers, and the works of medieval theologians to advance his own interpretations and practical spiritual advice in the text. He quotes a wide range of scripture and authors to offer what he calls a "radical remedy" for the disordered lust for riches that causes men to unjustly acquire wealth and hoard it (Osuna [1542] 2002, p. 879), relying particularly on Psalms, Proverbs, Ecclesiastes, Wisdom, Isaiah, Hosea, the Gospels of Matthew and Luke, Paul's letters, the Letter of James, Origen, Chrysostom, Augustine, and Jean Gerson.[5]

Drawing from his Franciscan predecessors, Osuna writes in the tradition of Francis of Assisi and Bonaventure regarding creation as a book to be read. He also cites Nicholas of Lyra in his discussion of the meaning of Francis of Assisi's stigmata (ibid., pp. 324–25). Osuna contributes especially to the textual tradition of friars of the Minor Observance, sharing with Scotus, Peter John Olivi, and Fransesc Eiximenis concerns, metaphors, and hermeneutic style. While Osuna draws from Scotus his notions of the importance of free will and the will as an intellectual power, *Consolation for the Poor and Warning for the Rich* particularly engages aspects of Franciscan tradition found in the texts of Olivi and Eiximenis on money, poverty, markets, and community (Evangelisti 2009; McClure 2019). More comparative work should be done to explore how Osuna's work relates to that of Olivi, whose thirteenth century theory of *usus pauper*, or "poor use", was controversial and highlighted the ways Franciscan vows of poverty were contradicted by common practices (McClure 2019, pp. 340, 351–52). Closer study may reveal a debt to Olivi's understanding of the will and poverty, although Osuna does not cite Olivi explicitly.

Regarding Eiximenis, whose late fourteenth century writings on contracts, money, and the role of merchants continued to be influential in the Iberian Peninsula in the sixteenth century, Osuna does not engage openly with him. Eiximenis's political theory defined "the inextricable union of the market and civic society" and was concerned with proving the importance of merchants (Evangelisti 2009, p. 407). While Osuna may have shared with Eiximenis a desire to protect the commons and discourage hoarding, their philosophies of money and profit diverged. Osuna depicts money as saltwater that increases unhealthy thirst and he denounces the voracious greed for gold from the Indies that can turn a temple into a cave of thieves, although he only mentions *indios* [Indians] once in a reference to innocent nudity (pp. 561–62, 523). He condemns warfare, blames kings for the bloodshed of war, and calls bodily freedom a principal form of true wealth intended for all human beings. He offers an objection to slavery and to enslaving Black Africans instead of employing local laborers (while also approving the evangelization of "infidels" and the enslaved), and he argues that people who cannot work and need a livelihood should be supported by householders with means (pp. 654, 859–61).

Of the poor, he says, there are three types of spiritual wayfarers: walkers who labor to take care of themselves and their household, runners who work to also benefit others beyond their own households, and flyers who, additionally, contemplate God's love so that it manifests in their hearts (ibid., pp. 264–65). As Pérez has affirmed in his study of what he calls Osuna's theology of history: "Mercy appears as the nucleus of his theology of history as well as the mystical theology of the Franciscan author" (Pérez García 2019b, p. 489). Pérez makes the case that *Consolation for the Poor and Warning for the Rich* is a text that gives a clear political, material, and embodied dimension to Osuna's earlier works' treatments of mercy for one's neighbor (ibid., pp. 499, 520).

At the same time, it is crucial to acknowledge that at the margins of Osuna's central conversation with Christians, Jews and Muslims occupy the space of Other. In the context of the Spanish Empire and sixteenth-century Christianity, Muslim "infidels" were the enemy and Jews were exiles upon whom blame for Jesus' death was heaped along with a litany of other accusations. Recent converts, or *conversos*, would have been among those with whom Osuna interacted. While he refers compassionately to enslaved Muslim individuals in his work, briefly, in the context of hoping to convert them to Christianity (Osuna [1542] 2002, p. 655), he says nothing about the Inquisition's persecution of converts. Similarly, he is silent about contemporary Sephardic Jews who had immigrated to Flanders in waves after their expulsions from Spain in 1492 and from Portugal in 1496. Osuna limits himself to speaking about Jews almost exclusively as they appear in the Bible, except for a brief condemnation of Christian soldiers' raping of Jewish and Muslim women in contemporary North African wars (p. 803). However, he does compare Queen Esther, who Emily Colbert Cairns argues is "a foundational figure for *conversos* throughout the diaspora" (Cairns 2018, p. 170), to the Virgin Mary (Osuna [1542] 2002, p. 407).

Regarding women, the category *la mujer* "woman/wife", appears negatively as a model of human disobedience and source of lust and blame in Osuna's works, and he criticizes friars who break their vows by comparing them to female prostitutes. However, he also praises "good" wives and widows as models for men, and individual historical and biblical women appear positively throughout his works, sometimes as central interlocutors and patrons (Bultman 2019, pp. 13–15). How Osuna interacts with non-Christian and female Others offers a view into the European cultural transition to modernity in the sixteenth century and how theologians dealt with their own claims to authority as men. He defends his identity as an Observant Franciscan friar while offering a reformist critique of clergy. At the same time, he breaks with emerging economic and political power relations in audible but limited ways, which, I propose, are meaningful for twenty-first century readers to consider.

### 3.3. Overview of Osuna's Text

The form of *Consolation for the Poor and Warning for the Rich* mirrors the conventional Christian view of a principal social class division of poor/rich in its combination of two treatises. The first treatise, *Consolation for the Poor*, comprises 112 chapters that advise a reader how to best survive and understand poverty's many distinct trials of body and spirit, urging him to "not give up the treasure of your free will to the power of a man without a brain" (p. 656).[6] The second treatise, *Warning for the Rich*, is more incisively written. It comprises 74 chapters on how rich men should view wealth, avarice, and generosity, and how to morally use their power in practical ways. Osuna places his treatise for the poor first, he explains, because being poor involves more danger, and he directs it to all Christians, not just mendicant friars or Roman Catholic "apostólicos", but to any poor Christian who is in need of consolation (p. 269). Thus, the first treatise is addressed to a combined group: specifically to the poor, meaning readers who have little or no material wealth, as well as to readers who are, as in the beatitudes of Matthew, "poor in spirit", meaning those who embrace forms of voluntary poverty.

Because he is familiar with a history of diverse theological positions, Osuna's definition of poverty is complex. He defines the ideal of voluntary poverty, being poor in spirit in

practice, as any person who "only wants licit earnings, and works for a living, without deceit or flattery" (p. 794).[7] In the second treatise, he addresses the rich in particular, urging them to become poor in spirit. He also includes the "patient poor" as intended readers of the second part, with the goal of bringing the two groups together in mutual insight so that, "the poor not be very poor and the rich not be very rich, since all excess is a vice" (p. 687).[8]

Studying *Consolation for the Poor and Warning for the Rich* as one cohesive work, rather than reading and analyzing the two treatises separately, makes sense because there is a continuity of imagery and advice between them. Although Osuna's overall goal is ostensibly not to promote social cleavage, he offers extended teaching suffused with perceptible outrage on the punishment of the miserly rich at the Last Judgment (Pérez García 2019b, pp. 515–21). The first part of the text sometimes rumbles with the potential for inspiring revolt, and he considers the Peasant's War of 1525 in Germany explicitly, attempting at moments to explain Lutheran perspectives (Osuna [1542] 2002, pp. 321–22, 803). Yet, the most striking difference between the two treatises is Osuna's attitude toward the growth of Renaissance humanist studies. He demonstrates interest and respect for empirical science and natural philosophy throughout. However, he worries about the proliferation of secular print books. In the first treatise he states that new humanist educational trends in literature and philosophy are fueled by ambition and idle curiosity, posing a great danger of error, so readers should not worry if they do not have access to them (ibid., pp. 382–83). In contrast, he pitches his second treatise to reach lay readers who are already immersed in humanist texts. He makes an effort to demonstrate humanist knowledge, punctuating his discussion with snippets of learning included to appeal to well-educated men interested in Greek. Osuna names Socrates and Plato's *Republic*, for example, to support his view that laws should alleviate material poverty and increase everyone's wealth (p. 864). But he pokes holes in humanist authority when he disagrees with the writings of Lucian, expressing strong disdain for facile advice that the poor ought to simply work harder to improve their lives (pp. 865–66).

## 4. Osuna's Imagery: The Natural World and Jacob's Ladder

### 4.1. Sun, Moon, Animals, the Elements, and Earth

To inspire the minds of his readers, Osuna describes the world's original state as "Ser y rico" [existing richly] with goods, fortune, and grace for all beings (p. 689). He uses visual imagery of the natural world to guide his readers, leading them to reason out the law of generosity in creation, which can be deduced through observation. Osuna states that the Sun, like God, gives its light and warmth at the proper time and unceasingly (p. 689). He continues: "There is more to the Sun: so that, while our half of the world is not enjoying him, he goes running to the other hemisphere, and on the other side gives himself very joyfully and takes pleasure in, and is happy, seeing that there is someone day and night who receives his gifts" (p. 689).[9] Osuna develops this image, positing that the rich, like the Sun, should give away their wealth, while poor human beings are like the Moon, reliant on the wealth of the Sun's light to shine (p. 719). Thus, he reasons, rich men should virtuously model their actions on the natural world, which is imbued with divine generous intent in constant and merciful circulation.

Osuna teaches readers to use their senses to see this circulation of generosity everywhere, written in the book of nature (p. 737). Animals are temperate, not seeking more than they need: "Ask, then, man, the animals and they will teach you the religion of mercy" (p. 737).[10] This compassionate religion of mutual exchange is patently obvious as "natural law" so we must not allow greed to erase it in the human heart:

> So that avarice would not erase this natural religion from your heart, God thought it good to inscribe it on all of his creatures; because, if nature, as it can be proved, abhors a void and does not consent to it, or permit it, why is it, do you think, except because a void impedes the influences and communications that exist between superior and inferior things, high and low? From this it follows that the

elements are symbolic, that is, they participate with one another in a quality, and they help one another.[11] (p. 737)

In this way, Osuna guides his reader to question and to observe material things interacting dialogically. By contemplating the four elements—fire, air, water, and earth—one can understand the flux between spiritual and material worlds. Rather than fixing light/spirit and heavy/body in opposition, he employs inversions and blending to loosen these binaries with circulation, interchange, and transformation.

At the level of the material world's constituent elements, there is mutual aid. For this reason, Osuna refuses to map onto the cosmos a hierarchical ranking of elements, or human beings, with permanent values assigned. He refuses to picture a Chain of Being, which was to become a dominant concept.[12] His choice, in keeping with his background in nominalism and Neoplatonic mysticism, is evident in how his imagery disrupts stable and pure categories.[13] Osuna inverts the conventional order of the elements consistently, and his nested metaphors move creatively into one another through the pages of *Consolation for the Poor and Warning for the Rich* without losing any of their many meanings. While fire and air may represent "higher" spirit, he raises the importance of the "lower" elements, earth and water, transforming them into imagery of the human body and the flow of mercy. Earth and water participate in Jesus' wounds, and Jesus Christ's embodiment produces a fountain that reaches paradise; meanwhile immaterial fire and air momentarily descend to participate in internal forces of sin, producing greed and pride (pp. 728–30).

Earth—the human body—is a metaphor for all people in Osuna's book, but then again she is our rich mother. He writes, "And the Earth, as mother, sustains and feeds us with her fruits, and like to children she gives us her wealth" (p. 737).[14] Osuna builds this play of "low" as "high" and "inferior" as "superior", to relate such paradoxical inversions to Jesus' parable in the Gospel of Luke about Lazarus, the kind beggar at the gate of the miserly rich man: Lazarus reaches paradise, while the rich man is damned. With these overlapping associations, Osuna meant to persuade Western Europeans that the responsibility of the rich was to generously circulate their material resources—following God's original intention for creation—rather than accumulate and gate them. The kindness of rich men should be like a river that flows across the land, beyond their own families and social circles, not a private well (p. 736).

In Osuna's view, human greed introduces an unnatural void and a cruel separation in the spiritual circuitry of nature. That being the case, he favors depicting touch and movement, in keeping with his rejection of stratification and fragmentation: Lazarus permits dogs to lick his wounds; Angels travel up and down; Jesus Christ descended into his humanity and ascended. Similarly, Osuna teaches, all human beings have an internal column of joy and sadness in our hearts that microcosmically mirrors the fluctuating conditions of the larger world.

Not only does our heart move with emotion, but our mind moves its attention as well: "You are so changeable that you have something at hand and, without concentrating on that, you turn your attention to another thing, and another" (p. 678).[15] Movement is as inherent in human interiority as it is to the elements in Osuna's text. Everything is subject to change, internally and externally, and meant to flow in mutual participation. Therefore, Osuna's moral and mystical theology places supreme importance on the power of free will, which he considers the most important form of true wealth (p. 692).

### 4.2. Jacob's Ladder

He explains free will as consisting of reason as its root and volition as its ability to move; it is an original freedom of understanding enjoyed by humans before the Fall (pp. 692–93). With this foundation, Osuna turns to the image of Jacob's dream ladder in Genesis to persuade his readers to use their free will in combination with love. My reading of Osuna's use of the ladder concurs with that of Pérez, who explains the importance of love in Osuna's approach to materiality and the body (Pérez García 2019b, pp. 499–502). Osuna notes the many interpretations that exist of Jacob's ladder, and states that, in his

opinion, God uses the ladder to show how to use generosity practically to gain spiritual salvation (Osuna [1542] 2002, pp. 725–26). For Osuna, the ladder proves that the purpose of material wealth, in accordance with natural law, is to create favorable conditions across society, thereby increasing one's own human virtue. He argues this, over eight chapters, through a detailed exposition of Jacob's ladder as a great circuit of generosity that allows spirit and matter to influence, communicate, and unite with one another.

The ladder's two rails are as vitally important for this circulation as are its steps. The following passage exemplifies Osuna's transmission of conventional exegesis on the topic, which he reworks for his audience:

> The two rails of this ladder are two considerations that he who gives alms should have. The rail on the right, the first one, is for you to think about how you are giving alms to God first of all. Do not look at the man you give to, rather the One for whom you are giving, who is God our Lord . . . Place the other rail, or foot, of this ladder close to yourself, and you should do this when you think about asking God for things greater than those you give to the poor in alms. Think that for God, of whom every day you ask many things, you yourself are poor. With what audacity can you say the prayer of the *Our Father*, that is full of requests, if you turn your eyes away, not wanting to see or hear the poor?[16] (pp. 726–27)

Osuna affirms every rich man is simply a poor human being in God's eyes and the rails of Jacob's ladder represent a vital loop of giving that must not be blocked.

Keeping the connective circuitry of mercy working unobstructed and inclusively along these rails is crucial. The stairs of the ladder, for their part, do not represent categories of humanity. Rather, each of the three stairs is an internal spiritual effect of practicing generosity. Over time, through one's own giving, he explains, a person rises to the first step, becoming more resistant to greed and morally sound. To rise to the second step requires patience; but after making the effort to sow many seeds of charity, one can eventually notice material wealth growing along with the goodwill of other people. Finally, rising to the third step of the ladder requires learning to advocate for the needs of others, in this way placating God's judgment and winning forgiveness before death comes (pp. 728–35). To reach this third step Osuna recommends imitating the Canaanite woman of Matthew's Gospel who is persistent in her approach to Jesus on behalf of her ill daughter (p. 734).

### 5. Osuna's Imagery: Dark-Skinned Women and Crowns

*5.1. Female Biblical Figures*

Referencing Augustine, Osuna writes that especially Dukes and Counts should treat their vassals with the love of the Canaanite mother who so tenaciously advocated for her daughter (pp. 734–35). He adds, since they receive good rents from the people on their land, they ought to anticipate their people's needs and really feel and respond to their suffering (p. 735). The Canaanite woman, ignored by Jesus as an outsider "dog" at first (Matt 15: 22–28), is not a particularly prestigious figure for Osuna to present as a spiritual model for the most powerful men in his society. Yet it is common for Osuna to use female biblical figures in his work—women characters and allegorical figures—as role models for contemporary men, rather than suggesting they imitate Jesus. He asks men to learn from female figures, be like them, internalize their virtuous qualities, and, ultimately, sometimes view themselves spiritually as feminine, like a spouse joined in a mystical relationship with Jesus Christ as their bridegroom.

In *Consolation for the Poor and Warning for the Rich*, Osuna combines this internalization of two genders with a focus on poverty's relationship to outsider status, dark-skinned women, and biblical crowns. Building on patristic interpretations, he focuses first on Zipporah as a figure for God's people. Starting with Moses' wife, identified as Ethiopian in the Vulgate Bible (Num 12: 1), Osuna encourages his readers to contemplate her as a real woman of their own social world with his choice of the non-biblical descriptive word "prieta", a Spanish term of the time meaning a woman of African ancestry nearly black in color: Moses "took a very dark-skinned [*prieta*] wife" (p. 275).[17] He goes on to note: "One

never reads that the holy prophet Moses was unhappy in any way with his wife, nor did he ever criticize her, nor libel her with ill-willed repudiation" (p. 276).[18] Osuna compares Moses' and his wife's mutual loyalty, and God's punishment of Aaron and Miriam for objecting to their union, with Jesus' full embrace of his human condition and absolute poverty, which caused some to ignore and persecute him: "If [Jesus] had been rich they would have made a place for him and listened to his doctrine and answers, but Ethiopian poverty was cause for them to belittle his prophecies and sermons" (p. 276).[19]

Osuna underlines that Moses' wife prefigures Jesus' human condition. His discussion suggests that women who look like her are to be included in the body of the faithful according to the authority of the Bible. A figure for the human soul that is poor in spirit, she is beautiful. Her opposite is the corruption of greed and exclusion. Osuna alludes to the famed verses on Solomon's dark-skinned beloved (Cant 1: 5–6) and cites Wisdom 8: 2–4 to exalt how she embodies grace, kindness, and wisdom. In fact, he affirms, she is an image of the beautiful cleanliness of poverty, while greed in the rich man is ugly:

> But she is in herself beautiful because avarice is so ugly, so much so that barely or never can one find a rich man without its stain, so it seems that beautiful cleanliness suits this bride of Christ our Lord, who is like a lily among thorns. If you, being so blind, misjudge the colors of this lady, that does not stop her from being graceful to the angels and very kind in God's eyes. Where with Solomon we can say: 'Her have I loved, and have sought her out from my youth, and have desired to take for my spouse, and I became a lover of her beauty. She glorifies her nobility by being conversant with God: yea, and the Lord of all things has loved her. For it is she that teaches the knowledge of God and is the chooser of his works.'[20] (p. 279)

Osuna figures the nobility of wisdom using an image of a beloved woman, while he allegorically associates Moses' and Solomon's devotion to their partners with Jesus' voluntary poverty; her worthiness flowers in his crown of thorns. Osuna then guides his reader to see these feminine figures as analogous to the Jewish heroine Esther, who had to hide her identity before becoming a queen who saved her people, citing Esther 2: 15–17:[21]

> Moreover, to Moses, to Christ, and to the good Christian, and devout clergyman, Ethiopian poverty does not seem to be anything but so beautiful that of her we may say, 'And as the time came orderly about, the day was at hand, when Esther was to go in to the king. But she sought not women's ornaments, but whatsoever Egeus the eunuch the keeper of the virgins had a mind, he gave her to adorn her. For she was marvelously lovely, and to the eyes of all she appeared incredibly agreeable and amiable.'[22] (p. 277)

Esther's bearing and generous manner wins the favor of the formidable king of Persia; by approaching him on behalf of her people, clothed by Egeus since she herself owned no adornments, she becomes a capable protector and administrator of justice.

## 5.2. Queens and Crowns

Following Osuna's choices of scriptural quotations, the reader is not to imagine himself as a patriarch or king, like Moses, Solomon, or Ahasuerus, but as a woman in a mystical relationship with God. Because Ahasuerus protects the Jewish people at Esther's request, Osuna argues, she is also comparable to Mary the Mother of God; no woman other than Mary has been more greatly favored than Esther, he writes (p. 407). Therefore, he argues, Esther's queenly crown and Jesus' crown of thorns resonate together and blend as symbols of spiritual nobility.

Ultimately, Osuna affirms, one should embrace Christian poverty as the queen of the beatitudes and the most important among them (pp. 534–39). The wayfarer who reaches a state of spiritual perfection will thus gain the ability to visualize their own human soul as an earthly woman, who, because of her effort to do works of justice, is united with God, and crowned with wisdom. The endpoint of this intense play of signs and symbols is a

new way of understanding the sign John sees in Revelation: the woman clothed in the Sun. In the night sky, with a resplendent starry crown, the woman clothed in the Sun represents Mary and the Mother Church, yes. But moreover, for Osuna, the sign additionally signifies all the biblical women he links together, and represents a spiritually wise embodied human being, perhaps even—if he is spiritually "perfect" enough—the reader himself (p. 535).

## 6. From Waste and Exclusion to Inclusive Freedom

### 6.1. Causes of Poverty: Finance and Slavery

Osuna guides his readers to contemplate the dignity of human beings, using symbolic figures anchored in the likeness of a dark-skinned woman and mapping her onto the cosmos as an aspect of a composite, spiritual queen. Such mappings, as Wynter explains, define a "specific criterion of being human", and naturalize a mode of existence as supernatural truth (Wynter 2003, p. 271). I am suggesting that Osuna works with and reflects here the dominant European "systemic stigmatization, social inferiorization, and dynamically produced material deprivation" of Black Africans in his time, following the terms of Wynter's argument (ibid., p. 267). Might his text show signs of resistance to such stratification and exclusion? His thought heightens the value of earth and the body, compresses hierarchical human social ranks, and insists on the potential for superior generosity on the part of the poor, exemplified elsewhere in his text by the widow's offering of two coins in Mark 12: 41–44 and a discussion of how the poor can help each other in imitation of Francis of Assisi (Osuna [1542] 2002, pp. 353, 533). He combines this with a moral critique of the economic and social practices that were rising in sixteenth-century European cities and courts in parallel with humanist education.

Near the end of *Consolation for the Poor and Warning for the Rich*, Osuna lists specific causes of poverty in his day, rebuking the rich for their annoyance with poor people. He views these causes as mostly new social developments for which they are responsible:

1. Rich Christians are not as charitable as they were in the time of the early Church;
2. The rich now prefer to be served by Black slaves in their households;
3. The rich are taking control of lands and removing people and villages to pasture animals;
4. The many wars of territorial expansion underway are theft and murder;
5. Delusional fantasies lead people to unlimited greed and spending what they do not have;
6. Laws do not protect the poor from being preyed upon by the rich;
7. The rich maintain an exclusive society in which they benefit only one another;
8. Fraternities of rich merchants are monopolizing trade;
9. The rich are taking for themselves resources intended for the poor;
10. The rich plunder raw materials and flood markets with foreign goods (858–69).

As a friar, Osuna witnessed these practices at close range in various locations through his interactions with people under his pastoral care. During his years in the Habsburg Netherlands, 1534–1536, Antwerp was a bustling trade center and the site of the bourse of Antwerp, an early commodity exchange rebuilt in 1531 that became a model for the development of stock exchanges in other locales (Silver 2012, p. 17). With the following personal report, Osuna offers his readers a view into his direct experience of the subtleties of marketplace deals and the tricky practice of using *cambios* [bills of exchange], a fundamental early instrument of international finance (Reinert and Fredona 2017, p. 9).

In this excerpt, Osuna also explains the hierarchical social networks that permit greed and corruption:

> So that you can see how difficult the snares of avarice are to recognize, let me tell you that when I was in Antwerp, that most solemn marketplace of Europe where there is greater trade than in Venice, the merchants, having much doubt about their dealings, and whether they were making excessive profits or not, agreed to write to the doctors in Paris, begging and paying them to inform them about what they call bills of exchange here, and to tell them whether they were lawful

or not. I myself read, many times, the response and signatures of the doctors, who did not know how to say yes or no, since they found those snares of avarice so hidden and hard to see that they did not know how to unravel them. And that is why they did not respond other than that it was advisable for the merchants to deal with their money, licitly, in some other way, and this was the response of men who could not see the snares well. That is why, not knowing how to untie them, they answered a question they were not asked.

Do not be amazed that madmen can ask more than wise men can answer, because I warn you that many merchants have been found in Flanders who have the ingenuity of devils in cheating, and such as these lay those snares that David complained about. While I was there, I often heard the merchants say that neither Scotus nor Saint Thomas reached the subtleties of Flemish markets and exchanges, and I conceded this because I saw in them the secret snares of the devil into which the rich make the people with whom they deal fall, and, unawares, their very souls are caught there, which happens so subtly that all the scholars in the world who would scrutinize those sins would faint in the inquiry.

Since, as with all questionable things that touch the health of the soul, a man is obliged to take the safest route, you could ask what is the reason those merchants permit these bills of exchange, when all wise men have great scruples about them being a mortal sin and requiring restitution. I shall reply to this with an answer that a gambling page gave to his confessor once, who, when asked why he did not stop gambling, replied: 'Sir, I play at times with the head waiter, and the head waiter plays with the butler, and the butler with the treasurer, and the treasurer with the count, and the count with the duke, and the duke with the king, to do well and be more successful'. To this the confessor responded: 'The main fault lies there, because, if the king did not play, then the others could abstain'. I mean to say that, if kings did not make excessive profits, then that vice would be remedied and punished and that snare would be cut; but understanding the king is involved, neither scholars dare to judge it bad nor justices punish it, and in this way everything remains hidden and they are still attributing it all to the secret snares of the devil.[23] (pp. 834–35)

This is Osuna's report of what he witnessed: early forms of capitalist finance that seemed to him like gambling.

As Reinert and Fredona observe, this period of the "'commercial revolution' saw a remarkable transformation of mercantile practices, practices by which merchants were able to create a global trade in both commodities and luxury goods and to thereby enrich and empower urban Europe" (Reinert and Fredona 2017, p. 3). Meanwhile, "the interest or profit from issuing such bills of exchange could be included (or perhaps better, given the usury prohibition, hidden) within the exchange rate, artificially raised in the lender's favor" (ibid., p. 9). The theologians of Paris were afraid to denounce abuses when kings were implicated in the profit motive despite theological precedent (Reinert and Fredona 2017, p. 20). They preferred to blame the devil, Osuna implies, despite the fact that they knew full well that human beings in positions of authority were responsible.

### 6.2. Causes of Poverty: Warfare and Slavery

In opposition to twentieth-century neoliberal capitalist theory, Osuna's Franciscan perspective is that greed is not good. To view life as a game of profit is a kind of madness, Osuna argues, citing Wisdom 15:12 (Osuna [1542] 2002, p. 794). Warfare is nothing more than robbery and murder committed at the top of the social hierarchy: "kings kill their vassals to protect their wealth, and send them to wars, not to defend the faith, but money" (ibid., p. 872).[24] Osuna objects to slavery on similar grounds, that it robs inherent dignity from human beings while spreading misery and hatred: "The slaves who lack freedom and are subjected have fallen greatly from the dignity of man, who was created to be a master,

and have been made into beasts, which were to be subject to man, and for this slaves are despised and men abhor coming into such a state" (p. 654).[25]

Osuna does not go farther in his criticism of slavery in this portion of the text before he finds a potential benefit in enslavement that is, in his opinion, access to a Christian education. Later he brings up slavery again, this time criticizing its effects on the local poor. In Osuna's view slavery breaks the traditional practice of mutual caring in the households of the rich, excluding other people's needs and rights from consideration: "nowadays the rich spend a lot on keeping dogs and hawks and Black people, so that they only want to be served by Blacks, and not by needy people in their village" (p. 859).[26] Osuna's concern here centers on local common people. He illustrates the point that no humans should be treated as expendable waste with an anecdote about the archbishop of Toledo, Alfonso Carrillo de Acuña (1412–1482), who famously opposed the authoritarian policies of the "Catholic Monarchs", Isabel and Ferdinand. When an administrator told Carrillo he did not need all of his employees because some were superfluous, the archbishop refused to get rid of any of them, stating: "Well, the ones that I need should stay, because I need them; and the other useless ones should stay too, because they need me" (p. 860).[27]

*6.3. Free Will, the Patriarchal Household, and the Circuit of Mercy*

Osuna's approach to questions of freedom and justice centers on free will. His Franciscan view is that there is no greater detriment to a human being than when freedom suffers, because with it the potential for wisdom is lost. Free will for Osuna is necessary for the inner creativity of the soul to operate. Reason, as the root of reflection, in harmony with volition that moves, produces wise understanding (pp. 692–93). On the other hand, greed and violence force apart the internal spiritual exchange between reason and volition inside human beings, alienating the human species from its own nature. In this internal creative freedom, Osuna locates the foundations of justice.

Osuna's concerns with justice and the central role of households as sources of love and material well-being in his day corresponds, five hundred years later, with twenty-first century analyses of "how we lost the ability to freely recreate ourselves by recreating our relations with one another", as Graeber and Wengrow (2021, p. 514) have put it. The madness of Western Man's overrepresentation of himself and indifference to others can be traced to a concept of the patriarchal household that, with the rise of humanist classical education, newly gave itself the liberty to subjugate, enslave, and treat other human beings as objects:

> The Roman Law conception of natural freedom is essentially based on the power of the individual (by implication, a male head of household) to dispose of his property as he sees fit. In Roman Law property isn't even exactly a right; since rights are negotiated with others and involve mutual obligations; it's simply power–the blunt reality that someone in possession of a thing can do anything he wants with it, except that which is limited 'by force or law'. This formulation has some peculiarities that jurists have struggled with ever since, as it implies freedom is essentially a state of primordial exception to the legal order. (Graeber and Wengrow 2021, p. 508)

The absolute liberty to dominate other human beings in the intimacy of the household—justified by patriarchal privilege—is utterly critical to "how we got stuck", Graeber and Wengrow affirm (ibid. p. 514).

This "primordial exception to the legal order" is embedded in white supremacist capitalist patriarchy; it extracts the free will of intimate others and breaks the loop of reciprocity. The natural circulation of generosity in what should be mutual bonds of community is lost to a violent forcing of "gifts" to flow in one direction only. This break of an ideal of reciprocal care within human communities that makes people objects to be used for profit, discarded, or erased, is, in Osuna's thought, a perverse profanation of God's mercy.

### 6.4. A Politics of Care

As Mikkel Krause Frantzen and Jens Bjering have recently argued, in the twenty-first century we are seeing the result of the breakage of mutual bonds of community in extreme examples of ecological toxicity, such as microplastics and global warming, which they term "the hyperabject", and which can be "defined as a planetary infrastructure of waste" (Frantzen and Bjering 2020, p. 89). Economic and ecological circulatory loops are clogged by what is "discarded, junked and excreted"; a waste that was intentionally produced is spreading dystopia (ibid., p. 89). Today the hushed externalities of infinite growth economics, integral to a monstrous necropolitics for which powerful nations and companies are disproportionately responsible, have taken the place of the "hidden snares of the devil" which Osuna recognized as a cover for the responsibility of powerful people in his own society.

Frantzen and Bjering call for an abolition of forcing human communities to integrate with toxic waste, suggesting instead "a politics of care" that prioritizes human agency (Frantzen and Bjering 2020, p. 105). Economist Raworth (2017) has already proposed such "care" in a new set of images and actions for recomposing human relationships anew in her *Doughnut Economics*. She replaces gross national product, and the one-directional line drawing of GDP that traces an impossible curve of infinite economic growth, with realistic measures of success centered on inclusive freedom and dignity. The Doughnut Economics Action Lab website explains the concept:

> The Doughnut consists of two concentric rings: a social foundation, to ensure that no one is left falling short on life's essentials, and an ecological ceiling, to ensure that humanity does not collectively overshoot the planetary boundaries that protect Earth's life-supporting systems. Between these two sets of boundaries lies a doughnut-shaped space that is both ecologically safe and socially just: a space in which humanity can thrive. (About Doughnut Economics 2022)

Raworth's diagram of the doughnut accounts for limits instead of erasing responsibility for human suffering. It is a figure of some complexity; as with the polyhedron its intricacies aim for regeneration.

## 7. Correspondences with *Fratelli Tutti*

Osuna's sixteenth-century thought interfaces with patriarchal and imperialist structures, sometimes as a condition for speaking, other times, arguably, because to do so suited his interests. His text contains more substance, and presents more questions, than I was able to explore here. Certainly he argued, near the end of his life, in *Consolation for the Poor and Warning for the Rich*, that the spiritual purpose of material wealth is to give it away generously to relieve the suffering of others.

Accepting Earth's finiteness as a limitation and boundary for one's own material desires has a foundation in scripture that he emphatically locates in Isaiah 5:8 and Hosea 5:10 (Osuna [1542] 2002, p. 347). Could his recognition of limits, along with his description of a natural circuit of generosity at the onset of colonial capitalist expansion, be seen as a forerunner of regenerative economic concepts? Could Osuna's nested and layered imagery provide a Franciscan precedent for engaging the complexity of a concept like Francis' polyhedron and for imagining an inclusive future world as a "beautiful polyhedral reality"?

Among the common points of intersection between *Fratelli Tutti* and *Consolation for the Poor and Warning for the Rich* are three strategies for action both texts recommend:

1. Distributing material resources while becoming content with possessing less;
2. Taking pride in local communities and cultures;
3. Overcoming delusions and indifference to the needs of others.

Francis and Osuna both argue for distributing material resources and against hoarding. Francis writes, "In the first Christian centuries, a number of thinkers developed a universal vision in their reflections on the common destination of created goods. This led them to

realize that if one person lacks what is necessary to live with dignity, it is because another person is detaining it" (Francis 2020, § 30). Osuna likewise writes:

> A common secular saying goes 'It is better to have extra than to run short of something.' However, the religion of which we speak, and any other, if it is good, says that it is better that we lack it, because what you lack makes up what another poor person needs, and another lacks what you have left over. Let no one take from kindness its condition, which is to flow forward. It does not love wells, but rivers.[28] (Osuna [1542] 2002, p. 736)

Both texts rely, as well, on the tradition of appreciating one's own land and caring for one's local community and people. Francis writes:

> The solution is not an openness that spurns its own richness. Just as there can be no dialogue with "others" without a sense of our own identity, so there can be no openness between peoples except on the basis of love for one's own land, one's own people, one's own cultural roots. I cannot truly encounter another unless I stand on firm foundations, for it is on the basis of these that I can accept the gift the other brings and in turn offer an authentic gift of my own. I can welcome others who are different, and value the unique contribution they have to make, only if I am firmly rooted in my own people and culture. Everyone loves and cares for his or her native land and village, just as they love and care for their home and are personally responsible for its upkeep. (Francis 2020, § 36)

Osuna too affirms human beings' ability to create local social cohesion, giving primacy of place to the capacity for contemplating and feeling love: "Love is a book, and a tree, and a treasure, and a precious gem, and everything else you might wish to think about, if you scrutinize its mysteries" (Osuna [1542] 2002, p. 500).[29]

A crucial weak spot in their common counsel seems, to me, clearly identifiable in the third item: the persistence of delusion and indifference. Those most responsible for creating poverty and sowing political division continue to deflect responsibility, hence the continued need for treatises and letters directed to them. Convincing rich readers to attempt to overcome their indifference to the needs of others and give up avarice is a staggering task. The delusion, as Francis explains, of "thinking that we are all-powerful, while failing to realize that we are all in the same boat" is pervasive (Francis 2020, § 8). After comparing Osuna's 1542 work with *Fratelli tutti* I was struck by how long this "failing to realize" has been ongoing, particular by those of us who subscribe to a static concept of Western Man, rather than imagining, following Wynter, that we are a whole human species with behavioral autonomy (Wynter 2003, p. 260).

In the last sentences of Osuna's text, a boat also appears. He has his reader imagine himself sailing on the ocean in a merchant vessel. He entreats him—pleads with him—to start giving away his excess wealth to those in need now, like a man seeing an approaching storm and throwing off gold cargo, if only to ensure his own survival (Osuna [1542] 2002, p. 891). How to convince the heirs of patriarchal power to overcome their indifference? A good start is listening to the voices of women; better yet would be to break open the bibliography of encyclicals to be inclusive of long-standing scholarly perspectives critical of the legacies of patriarchy and colonialism.

**Funding:** This research received no external funding.

**Data Availability Statement:** Not applicable.

**Acknowledgments:** I would like to thank Julia Hernández for reading and commenting on an early version of this essay and the readers for their insights.

**Conflicts of Interest:** The author declares no conflict of interest.

## Notes

1.   I use Mariano Quirós García's 2002 edition. The full title in Spanish is "*Quinta parte del Abecedario Espiritual*, de nuevo compuesto por el padre fray Francisco de Osuna, que es *Consuelo de pobres y Aviso de ricos*. No menos útil para los frayles que para los seculares y aun para los predicadores. Cuyo intento deve ser retraer los hombres del amor de las riquezas falsas y hazerlos pobres de espíritu." [*Fifth Part of the Spiritual Alphabet*, newly composed by Father Friar Francisco de Osuna, called *Consolation for the Poor and Warning for the Rich*. No less useful for friars than for laymen and even for preachers. Whose intent should be to draw men away from the love of false riches and make them poor in spirit.] (Osuna [1542] 2002, p. 231). There is no English translation of the work, so I translate quotations from Osuna's text into English and offer the Spanish original in the notes.

2.   Thomas Merton favorably reviewed the earliest English Translation of *The Third Spiritual Alphabet* in *The Commonweal* in 1948 (Merton 1948). Laura Calvert's comprehensive study of Osuna's six-volume *Spiritual Alphabet* with a focus on his mystical language spearheaded the growth of scholarship in English on Osuna (Calvert 1973). Mary E. Giles' edition and translation of the first published work of his career, *The Third Spiritual Alphabet* (Giles 1981), provided a major advance for knowledge of Osuna in the Anglophone world. Jessica Boon offers an in-depth study of the mystical method of Bernardino de Laredo's Franciscan recollection in Osuna's era (Boon 2012). Dale Shuger analyzes Osuna's mystical discourse within the context of mysticism in Early Modern Spain (Shuger 2022, pp. 25–41). For more on Osuna and recollection see Whitehill (2007) and Carolyn Medine's recent comparison of Osuna's meditative method with Buddhist meditation (Medine 2021). For a general overview of scholarship on Osuna and updated biography in English that draws from recent Spanish sources, see Bultman (2019, pp. 1–18). I differ from some scholars in my view that Osuna intended to include the poor, women, and recent converts in the practice of recollection, based on ample evidence in his *Norte de los estados*, sermons, and *Consolation for the Poor and Warning for the Rich*.

3.   See King's discussion of "fungibility" in her chapter 5, "Ceremony for Sycorax" (King 2019, pp. 175–206). See also Boon's provocative discussion of monetary exchange value and *conquistador* mentality in a passage from the first part of Osuna's *Spiritual Alphabet* and the sermons of Franciscan abbess Juana de la Cruz (1481–1534) (Boon 2015, pp. 395–96). Boon argues that "early modern Castilian mystical texts rely on spiritual negotiation and embodied commerce to delineate what God has to offer souls and what souls have to offer God" (p. 396). I concur with Pérez, in his study of Osuna's "theology of history" (Pérez García 2019b, p. 489), and with Boon that for Osuna "the economy, the body, and salvation are intimately linked" (Boon 2015, p. 395).

4.   Foundational studies of Osuna's mysticism and his brush with *alumbrado* [illuminist] persecution are Andrés (1975, pp. 107–67), Hamilton (1992), and López Santidrián (2005, pp. 16–19). As he wrote, Osuna would have been aware of his relatively low class position and association with controversial or weak patrons. These factors, together with his reformist views, interest in Lutheranism, association with peers accused of *alumbradismo* in the 1520s, status as a teacher who wrote in the vernacular for common people, and his criticism of clergy who did not adhere to vows of poverty, put him at some risk for persecution and likely played a role in his decision to leave Spain.

5.   Osuna states his remedy is radical, "este remedio es radical" to mean that it is intended to pull greed from the human heart by its root (Osuna [1542] 2002, p. 879). His six-chapter discussion of how to heal what he considers the "nearly incurable illness of avarice" comes at the end of *Warning for the Rich* (ibid., pp. 875–91).

6.   Translations to English are mine; "el thesoro de tu libre alvedrío no lo renuncies en poder de hombre sin seso" (Osuna [1542] 2002, p. 656).

7.   "no quiere ganancia, sino lícita, y gana de comer trabajando, sin trampas ni lisonjas" (Osuna [1542] 2002, p. 794).

8.   "ni el pobre sea muy pobre ni el rico sea muy rico, que toda demasía es viciosa" (Osuna [1542] 2002, p. 687).

9.   "Tiene más el sol: que, mientras nuestro medio mundo no goza d'él, se va corriendo al otro emisperio, y en la otra parte se da muy alegre y se goza y alegra viendo que ay quien de día y de noche resciba sus dones" (Osuna [1542] 2002, p. 689).

10.   "Pregunta, pues, hombre, a los animales y enseñarte an la religión de la misericordia" (Osuna [1542] 2002, p. 737).

11.   Osuna uses the term "ley natural" [natural law] in a passage just preceding this quotation about natural religion ([1542] 2002, p. 737); "Porque la codicia no borrase en tu coraçón esta religión natural, tuvo Dios por bien de la escrevir en todas sus criaturas; porque, si la naturaleza, según se prueva, aborresce la vacuydad y no la consiente, ni permite, ¿por qué, si piensas, es, sino porque la vacuydad impide los influxos y comunicaciones que ay entre las cosas superiores e inferiores, altas y baxas? De aquí es que los elementos son simbolizantes, esto es, participan unos con otros en una calidad, y se ayudan unos a otros..." (ibid., p. 737).

12.   For analysis of the concept of the "Great" Chain of Being and how it provided a justification for rigid and violent social hierarchy in the Spanish Empire from the fifteenth century, see Wynter (2003, pp. 274, 296–303). For the ways the Chain of Being bestialized enslaved people and was used by Christian abolitionists, see Jackson (2020, pp. 48–55).

13.   My understanding of Osuna's Neoplatonic mysticism and discussion of his reading of Jacob's ladder is indebted to Miles (1999).

14.   "Y la tierra, como madre, nos sustenta y mantiene con sus fructos, y como a hijos nos da sus riquezas" (Osuna [1542] 2002, p. 737).

15.   "Eres tan mudable que tienes entre manos una cosa y, sin parar mientes en aquélla, entiendes en otra y en otra" (Osuna [1542] 2002, p. 678).

16 "Los dos braços de aquesta escalera son dos consideraciones que ha de tener el que haze limosna. El braço diestro y primero es que pienses cómo la das a Dios primeramente. No mires al hombre, sino a Aquél por quien se la das, que es Dios nuestro Señor . . . El otro pie o braço d'esta escalera pon cerca de ti y hazerlo as quando pensares que tú as de pedir mayores cosas a Dios, al qual das en el pobre limosna. Piensa que para con Dios tú mesmo eres pobre, al qual cada día pides muchas cosas. ¿Con qué ossadía podrás dezir la oración del *Pater Noster*, que está llena de peticiones, si buelves tú los ojos no queriendo ver ni oýr al pobre?" (Osuna [1542] 2002, pp. 726–27).

17 "tomó una muger prieta" (Osuna [1542] 2002, p. 275).

18 "No se lee que'el sancto profecta Moysén tuviesse algún descontento de su muger, ni jamás la vituperasse, ni le diesse por malquerencia libelo de repudio" (Osuna [1542] 2002, p. 276).

19 "Si fuera rico diéronle lugar y oyeran su doctrina y respuestas, empero la pobreza etiopisa fue causa que menospreciassen sus profecías y sermones" (Osuna [1542] 2002, p. 276).

20 "Empero, es en sí hermosa porque, siendo tan fea el avaricia, que apenas o nunca se halle un rico sin mácula, bien parece que la hermosa limpieza conviene a esta esposa de Christo nuestro Señor, que es como lirio entre espinas. Si tú, como seas ciego, juzgas mal de las colores d'esta señora, no por esso dexa de ser a los ángeles graciosa y a Dios muy amable. Donde con Solomón podemos dezir: 'Ha ésta amé y acabé de buscar dende mi juventud, y procuré de la tomar por esposa. Y soy hecho amador de su hermosura, cuya generosidad glorifica el que tiene compañía de Dios, porqu'el Senor de todas las cosas la amó, que enseñadora es de la disciplina de Dios y electora de sus obras'". (Osuna [1542] 2002, p. 279).

21 See Cairns' study of the meaning of Esther, as queen, in the sixteenth-century Spanish Empire and the Sephardic diaspora. Her argument that "Esther is the patron-saint for crypto-Jews and *conversos* in the early modern period" provides context for considering the wider implications of Osuna's use of Esther here (Cairns 2018, p. 1).

22 "mas a Moisén, ni a Christo, ni al buen christiano, ni al devoto religioso, no parece la pobreza etiopisa, sino tan hermosa que d'ella podamos dezir: 'Bolviéndose por orden el tiempo, instava ya el día en que Hester devía entrar al rey, la qual no buscó mugeril atavío; mas Egeo, que guardava las vírgines, le dio las cosas que él quiso para su ornamento, porque era hermosa a maravilla y a los ojos de todos parecía graciosa increyblemente y amable' . . . " (Osuna [1542] 2002, p. 277).

23 "Porque veas tú quán malos son de ver los lazos de la avaricia, te hago saber que, estando yo en Enveres, que es aquella solempníssima tienda de Europa donde ay mayor trato que en Venecia, los mercaderes, teniendo mucha duda en sus tratos, si eran logros o no, acordaron de embiar a los doctores de París, rogándoles y pagándoles porque les informassen de los que allá se llaman cambios, y les respondiessen si eran lícitos o no. Yo mesmo leý, muchas vezes la respuesta y firmas de los doctores, que ni supieron dezir sí ni no, de manera que hallaron tan ocultos aquellos lazos del avaricia que no supieron desatallos. Y por esso no respondieron sino que les era de aconsejar que tratassen con su dinero en otra cosa lícitamente, y esta repuesta es de hombres que no pudieron ver bien los lazos. Y por esto, no sabiendo desatallos, respondieron lo que no les preguntavan. Tú no te maravilles que los locos puedan preguntar más que los sabios responder, porque te aviso que se an hallado en Flandes muchos mercaderes que tienen ingenio de diablos en trampear, y estos tales hazen aquellos lazos de que se quexava David. Yo, estando allá, oýa muchas vezes dezir a los mercaderes que ni Escoto ni sancto Thomás alcançaron las sotilezas de la mercaduría y cambios flandescos, y esto concedíalo yo porque veýa en ellos los secretos lazos del demonio en que los ricos hazen caer a las personas con quien tratan y, no parando mientes, prenden allí sus mesmas ánimas, lo qual acaesce tan subtilmente que todos los letrados del mundo que escudriñassen aquellos pecados desfallecerían en la inquisición. Como en las cosas dudosas que tocan a la salud del ánima sea hombre obligado a tomar lo más seguro, podrías preguntar qué es la causa por que aquellos mercaderes no dexan aquellos cambios, pues todos los sabios tienen grande escrúpulo que son pecado mortal y traen obligación a restituyr. Podríate yo responder a esto una respuesta que dio un paje jugador a su confessor, el qual, como le dixessen por qué no dexaba el juego, respondió: 'Señor, yo juego a las vezes con el mastresala, y el mastresala juega con el mayordomo, y el mayordomo con el thesorero, y el thesorero con el conde, y el conde con el duque, y el duque con el rey, por medrar y más valer'. A esto respondió el confessor: 'La culpa principal está aý, porque, si el rey no jugasse, luego se podían abstener los otros'. D'esta manera digo yo que, si los reyes no tomassen a logro, luego se remediaría y castigaría aquel vicio y se cortaría aquel lazo; mas entendiendo el rey en ellos, ni los letrados osan determinallo por péssimo ni las justicias castigallo, sino que assí se quede todo solapado y se llamen todavía lazos ocultos del demonio malo" (Osuna [1542] 2002, pp. 834–35).

24 "los reyes matan vasallos por defender sus riquezas, y embíanlos a las guerras, no por defender la fe, sino los dineros" (Osuna [1542] 2002, p. 872).

25 "Los esclavos que carescen de libertad y están subjectos mucho an caýdo de la dignidad del hombre, que fue criado para señor, y se an conformado con las bestias, que fueron al hombre subjectas, y por esto los esclavos son menospreciados y aborresce hombre venir en tal estado" (Osuna [1542] 2002, p. 654).

26 "ya los ricos gastan mucho en mantener perros y halcones y negros, de manera que no se quieren servir sino de negros, y no de menesterosos de su pueblo" (Osuna [1542] 2002, p. 859).

27 "Pues, éstos que yo he menester quédese, porque los he menester; y estos otros inútiles quédense también, porque ellos me an menester a mí" (Osuna [1542] 2002, p. 860).

28      "Conclusión averiguada es acerca de los seglares: 'Más vale que sobre que no falte'; empero, la religión de que hablamos y qualquier otra, si buena es, no dize sino que es mejor que nos falte, porque a otro pobre remedia lo que a ti te falta, y a otro falta lo que a ti sobra. Pues, ninguno quite a la bondad su condición, que es passar adelante. No ama los pozos, sino los ríos" (Osuna [1542] 2002, p. 736).

29      "El amor es libro, y árbol, y thesoro y piedra preciosa, y todo lo que más quisieres pensar, si escudriñas sus misterios". (Osuna [1542] 2002, p. 500).

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
