# Peer review of "Waste, Exclusion, and the Responsibility of the Rich: A Franciscan Critique of Early Capitalist Europe"

_religions, doi:10.3390/rel13090818_

Round 1
Reviewer 1 Report
This is a wonderful contribution to broader discussions about historical arguments about social divisions and their relevance to contemporary concerns about a fragmented social order. The attention to decolonial, feminist, and racial concerns is very fine, and a helpful framing to this scholarly project. Author is to be commended for sensitive retrieval of the Franciscan social and economic philosophy of Francisco de Osuna, and for bringing this in to dialogue with Fratelli Tutti by Pope Francis. Author is to be commended for calling our attention to this part of the Spiritual Alphabet and bringing it to the attention of the English-speaking scholarly world.
The interpretative frame provided is robust, with careful attention to Biblical and patristic precedent. However, author does not fully contextualize Francisco de Osuna as a member of the Franciscan intellectual tradition. The concerns, metaphors and hermeneutic style draw generously from Scotus, Peter of John Olivi and Francesc Eiximenis. The latter in particular addressed questions of economic philosophy and political economy on the Iberian Peninsula. (See Paolo Evangelisti, “Contract and Theft: Two Legal Principles Fundamental to the civilitas and res publica in the Political Writings of Francesc Eiximenis, Franciscan Friar,” Franciscan Studies 67, 2009, 405-426.) It may be that space limitations do not permit a proper contextualization, but author might gainfully consider drawing out some more specific preoccupations and methodological frames from these representatives of Franciscan intellectual tradition for greater religious and historical context, if for no other reason than that fact this is identified in the introduction and "Franciscans" is a keyword.
Author is likewise to be commended for articulating Francisco de Osuna's work with that of Pope Francis in Fratelli Tutti. For those who know this encyclical, the parallels are immediately relevant, however, author does not fully justify why this text, among many other contemporary texts, be chosen. Author might gainfully state more clearly the rationale for this (as opposed to, say, Laudato Si or the UN Sustainable Development Goals).
This scholarly article is insightful, original, and thought-provoking. Author is to be commended for breadth of understanding across fields and the ability to recognize and articulate patterns that advance multiple fields of scholarship.
Reviewer 2 Report
This is an original, insightful and critical essay. The author writes clearly and concretely, taking the reader with her in a logical argument. It demonstrates sound scholarship.
The author has shown indirectly the Franciscan sources of Osuna's thought (the importance of free will, the will as an intellectual power (Scotus), the creation as a book to be read (Francis of Assisi, Bonaventure), etc. Osuna's economic theory as presented by the author draws on the early Franciscan understanding of poverty and fraternity, and poor use as developed by Peter of John Olivi).
The author makes a novel, credible and explicit connection between the economic theory of Pope Francis in Fratelli tutti and Osuna's text.
